# Ability of Countermovement Jumps to Detect Bilateral Asymmetry in Hip and Knee Strength in Elite Youth Soccer Players

**DOI:** 10.3390/sports11040077

**Published:** 2023-03-31

**Authors:** Hailey L. Wrona, Ryan Zerega, Victoria G. King, Charles R. Reiter, Susan Odum, Devon Manifold, Karyn Latorre, Timothy C. Sell

**Affiliations:** 1Department of Biomedical Engineering, University of North Carolina Chapel Hill, Chapel Hill, NC 27514, USA; 2Atrium Health Musculoskeletal Institute, Charlotte, NC 28207, USA; 3Physical Therapy Division, Duke University School of Medicine, Durham, NC 27710, USA; 4Charlotte Football Club, Charlotte, NC 28202, USA

**Keywords:** football, injury prevention, biomechanics, risk factor, team sport

## Abstract

Clinicians frequently assess asymmetry in strength, flexibility, and performance characteristics as a method of screening for potential musculoskeletal injury. The identification of asymmetry in countermovement jumps may be an ideal method to reveal asymmetry in other lower extremity characteristics such as strength that otherwise may require additional testing, potentially reducing the time and burden on both the athlete and clinicians. The present study aims to examine the ability of asymmetry in both the single-leg and two-leg countermovement jump tests to accurately detect hip abduction, hip adduction, and eccentric hamstring strength asymmetry. Fifty-eight young male elite soccer players from the same professional academy performed a full battery of functional performance tests which included an assessment of hip adductor and abductor strength profiles, eccentric hamstring strength profiles, and neuromuscular performance and asymmetries during countermovement jumps. Bilateral variables attained from both the single-leg and two-leg countermovement jump tests included concentric impulse (Ns), eccentric mean force (N), and concentric mean force (N) computed by the VALD ForceDecks software. Average maximal force (N) was calculated bilaterally for the strength assessments. Asymmetry was calculated for each variable using 100 × |(right leg − left leg)/(right leg)| and grouped into three categories: 0 to <10%, 10% to <20%, and 20% or greater. Analyses were performed for the two higher asymmetry groups. The accuracy to detect strength asymmetry was assessed as the sensitivity, specificity, and predictive values for positive and negative tests. The outcomes from the accuracy assessments suggest that the single-leg countermovement jump concentric impulse variable at the 20% threshold is indicative of a youth male soccer player having hip adduction strength asymmetry while also demonstrating more accuracy and applicability than the two-leg countermovement jump concentric impulse variable.

## 1. Introduction

Over the last three decades, there has been a significant increase in youth soccer program participation in the United States [1,2]. Unfortunately, soccer participation is associated with a higher incidence of injury compared to other organized team sports such as basketball, volleyball, and baseball [3,4], with elite youth soccer players enduring an average of 2 to 19.4 injuries per 1000 h of exposure [5]. A survey of youth soccer players between the ages of 14 and 16 years over a ten-season period discovered that lower extremity injuries account for 78.1% of reported injuries while muscle strains of the hamstring, psoas, and thigh regions accounted for 15.3% of these lower extremity injuries [6]. The increased participation in youth soccer and the frequent occurrence of injuries presents coaches, trainers, and clinicians with a paramount duty to develop accurate, effective, and efficient injury prevention screening methods.

Clinicians, coaches, and trainers frequently assess bilateral asymmetries in strength, flexibility, and performance as a method of screening for potential injury. The measurement of countermovement jumps (CMJs) on a dual force plate system allows for the assessment of an athlete’s performance and asymmetries during a relatively universal athletic movement [7,8]. For instance, Raya-González et al. concluded that CMJ testing revealed higher magnitudes of asymmetry values, and consequently, a heightened potential injury risk as compared to a battery of other jumping, change-of-direction, and range-of-motion tests in youth female soccer players [9]. Muscle strength asymmetry assessments of the hip adductors, hip abductors, hamstrings, and quadriceps have also shown the ability to predict injury occurrence [10,11,12]. In another example, Dauty et al. demonstrated that bilateral differences in isokinetic hamstring strength predicted injury status in professional soccer players [13]. The presence of asymmetries can reveal unilateral strength deficits and, when left untreated, muscle deficits can result in athletes presenting more than four-fold the risk of injury [14].

Beyond soccer-based studies, additional previous research involving youth handball [15,16], tennis [17], basketball [16], and volleyball [16] reported interlimb asymmetries were associated with reduced sprint speed, change of direction speed, jump performance, and overall athletic performance. Additionally, the asymmetries appeared to be direction-specific, often linked to the weaker leg presenting the potential opportunity to train and improve the weaker limb during sport-specific skills [16,18].

The examination of eccentric hamstring, hip adductor, and hip abductor strength metrics and asymmetry will provide scientific evidence of a large cohort of youth soccer players’ lower extremity strength metrics which have not been established with relation to injury prevention [19]. Additionally, the evaluation of the relationship between the more commonly utilized CMJ asymmetry measures and lower extremity strength asymmetries may be an ideal method to prevent the need for an additional battery of tests when performing injury prevention screening. The present study aims to examine hip and eccentric hamstring strength and present asymmetries in a population of elite youth athletes. A secondary goal of the study is to determine the ability of asymmetry in both the 2L-CMJ and 1L-CMJ tests to accurately detect hip abduction, hip adduction, and eccentric hamstring strength asymmetry. The use of a CMJ asymmetry test as a link to other external strength indicators would streamline early detection of weakness in an athlete that could potentially lead to injury. The CMJ test would act as a step in the injury prevention process to monitor athletes and allow the clinicians to target specific muscle groups.

## 2. Materials and Methods

### 2.1. Subjects and Study Design

This was a cross-sectional study design. The Strengthening the Reporting of Observational studies in Epidemiology (STROBE) guidelines were followed in the preparation of this paper and reporting of results. This study was approved by the Atrium Health Institutional Review Board and all participants and parents or guardians completed written informed assent and consent, respectively.

Participants were rostered on the under-14, under-15, and under-17 teams of a professional youth soccer academy who completed testing on one day in July 2021, which coincided with the beginning of their pre-season training. Although the practices can vary in intensity, they players often train two to three hours each day. Normal season schedules for the academy include training four days per week and match play on the weekends. Players had not been previously exposed to team-based injury prevention programs to target muscle strengthening. Athletes were only excluded if not cleared by medical personnel to participate in the tests outlined in the study.

### 2.2. Procedures

Players were asked to complete a demographics questionnaire including date-of-birth, height, weight, position, and whether they were fully cleared to play. Clearance to play was based on approval from the team physician or certified athletic trainer. Testing was performed prior to practice, on-site, or as part of a pre-season screening. The testing was organized by the youth soccer academy staff and Atrium Health leadership and the testing was completed by qualified Atrium Health biomechanical staff. There was no warm-up requirement of any of the participants; however, each participant was required to perform a warm-up trial to ensure proper placement prior to data collection.

A full battery of functional performance tests followed, which included an assessment of hip adductor and abductor strength profiles, eccentric hamstring strength profiles, and neuromuscular performance and asymmetries during countermovement jumps. The order of completion of the performance tests was not randomized but instead based on the availability of the participants. The testing sessions were facilitated in a station-like manner, with each participant moving from one station to the next in a standardized order.

#### 2.2.1. Hip Strength Testing

Each participant completed adductor and abductor tests using a fixed dynamometer (ForceFrame, VALD, Brisbane, Australia), eccentric hamstring tests using the NordBord system (NordBord, VALD, Brisbane, Australia), and 1L-CMJs and 2L-CMJs on force plates (ForceDecks, VALD, Brisbane, Australia). The hip adductor and abductor testing positions with the fixed dynamometer consisted of the participant laying supine with hips flexed to 60 degrees, the femoral epicondyles centered on the dynamometer pads, and feet flat on the floor. The ForceFrame testing system allows for objective, repeatable, and reliable measurements with respect to isometric hip strength testing [20,21]. The participant was instructed to perform a 5 s maximal contraction followed by a 5 s rest period. The participant alternated between adduction and abduction trials until three maximal effort repetitions were performed for each action. The maximal force values of the three repetitions were averaged to yield the final force variables for analysis.

#### 2.2.2. Hamstring Strength Testing

The NordBord eccentric hamstring testing system provides an objective, repeatable, and reliable method to perform eccentric hamstring testing in the youth athlete participant group [22,23]. The eccentric hamstring testing position with the NordBord consisted of the participant placing the ankle hooks around the posterior ankle and kneeling on the NordBord with the head up and chest tall. The instructor informed the participant to uniformly and gradually lean forward while resisting the movement with the lower limbs. The participant was reminded throughout the testing session to keep their trunk and hips in a neutral position. Once the participant reached their maximal resistance position, they were then instructed to catch themselves on the ground and to crawl with their hands back to a kneeling position. The final force variables were calculated as the average of the max forces in the three repetitions, measured for each leg separately.

#### 2.2.3. Countermovement Jump Testing

Countermovement jumps are historically utilized in asymmetry research investigations and are repeatable and reliable testing methods within the youth athlete participant group [24,25,26]. The 2L-CMJ position with the ForceDecks consisted of the participant starting in a normal standing posture. The participant stood with their hands on their hips, chest up, and looking forward. The participant was instructed to remain completely still for 2–3 s before and between each jump. The instructor then directed the participant to perform a two-leg jump with the goal of jumping as high as possible. This procedure was repeated three times. The 1L-CMJ testing position was similar to the 2L-CMJ and performed similar except on one leg. Again, the participant began with their hands on their hip, chest up, and looking forward. The participant was instructed to remain completely still for 2–3 s before and between each jump. The instructor then directed the participant to perform a one-leg jump with the goal of jumping as high as possible. This procedure was repeated three times on each leg. The outcome variables attained from both the 1L-CMJs and 2L-CMJs include concentric impulse (Ns), eccentric mean force (N), and concentric mean force (N) computed by the VALD ForceDecks software. The final mean force variables used for analysis were the calculated average of the value for the three performed jumps.

### 2.3. Statistical Analysis

Statistical analyses were conducted using SAS/STAT software^©^, Version 9.4 for Microsoft Windows (SAS Institute Inc., Cary, NC, USA). Subject demographic data (height, weight, age) were collected for all participants. Normality of all numeric data was assessed visually with quantile–quantile plots and statistically with Shapiro–Wilk tests. Based on the distribution of the data, appropriate central tendency measures (mean, median) and variance measures (standard deviation (SD), range, interquartile range (IQR)) were calculated. Asymmetry was calculated for each variable using 100 × |(right leg − left leg)/(right leg)| [27]. Previous research suggests that asymmetry variable thresholds of 10–15% indicate the presence of abnormal differences between limbs and potential increased injury risk [28,29,30,31,32]. Symmetry was calculated and binned into 3 categories: 0 to <10%, 10% to <20%, and 20% or greater. Analyses were performed for the two higher asymmetry groups. The accuracy to detect strength asymmetry was assessed as the sensitivity (se), the specificity (sp), and the predictive values for positive (ppv) and negative tests (npv) with the measured CMJ variables defined as the new test and the measured strength variables defined as the gold standard. Sensitivity is the ability of a test to correctly classify, or rule-in, true positives in a population and is calculated as true positive/(true positive + false negative). Specificity is the ability of a test to correctly classify, or rule-out, true negatives in a population and is calculated as true negative/(true negative + false positive). PPV is the percentage of patients with a positive test response who actually result in a true positive case calculated using true positive/(true positive + false positive). NPV is the percentage of patients with a negative test response who actually result in a true negative case calculated with npv = true negative/(true negative +false negative). Higher positive and negative predictive values suggest that the new test is performing as accurately as the ‘gold standard’ [33]. In order to clarify the statistical values used throughout the analysis, the sensitivity and specificity can be considered to indicate the effectiveness of a test with respect to a trusted “gold standard”, while PPV and NPV can be considered to indicate the effectiveness of a test for categorizing people as having or not having a target condition [34].

The true positive (TP), false positive (FP), true negative (TN), and false negative (FN) counts were calculated to classify the similarities and differences between the jump asymmetry predictor, or test, variables and the strength outcome asymmetry variables. Ten and twenty percent were used as the defined asymmetry thresholds throughout the study. TP was assigned when both the evaluated jump and strength variable had an asymmetry value reaching the defined threshold. FP was assigned when only the jump variable had an asymmetry value reaching the defined threshold. TN was assigned when neither of the evaluated jump and strength variables had an asymmetry value reaching the defined threshold. FN was assigned when only the strength variable had an asymmetry value reaching the defined threshold.

## 3. Results

Fifty-eight elite youth male academy soccer players participated in the data collection session(s), with five players excluded due to missing or incomplete data. Therefore, 53 participants at an average age of 14.2 years were included in this study and all completed the testing activities, including 2L-CMJs, 1L-CMJs, and strength testing (Table 1). The descriptive characteristics of each strength variable measure are displayed in Table 2.

Additionally, each asymmetry measure is presented in Table 3. The 1L-CMJ EMF variable was excluded from the analysis due to zero participants reaching either asymmetry threshold.

The 1L-CMJ concentric mean force (CMF) variable had the greatest number of false negatives and 2L-CMJ eccentric mean force (EMF) had the greatest number of false positives with respect to all strength asymmetry variables at both the ten percent and twenty percent thresholds (Table 4 and Table 5).

The ability of the CMJ to accurately detect strength asymmetries varied across the strength variables and the defined asymmetry thresholds of either ten percent (Table 6) or twenty percent (Table 7). At the 10% threshold, the 1L-CMJ CMF asymmetry variable correctly classified asymmetry 86% of the time (e.g., specificity) with the hip adductor and abductor asymmetry variables defined as the gold standard (Table 6). The 2L-CMJ EMF asymmetry variable accurately ruled out asymmetry (e.g., sensitivity) in 63.6% of the tests with respect to the hip adductor asymmetry variable (Table 6). At the 20% threshold, the 1L-CMJ CMF asymmetry variable most accurately classified asymmetry with a specificity value of 98.1% with the hip abductor asymmetry variable as the gold standard (Table 7). At the 20% threshold, the 1L-CMJ concentric impulse (CI) asymmetry variable accurately ruled out asymmetry in 66.7%) of the tests with the gold standard of hip adductor asymmetry (Table 7). Throughout Table 5 and Table 6, both zeros and N/A values can be seen representing the sensitivity outcome variable. The zero value is a result of there being no true positives for the variable relationship. For example, the 2L-CMJ CI asymmetry variable and hip adduction 20% threshold sensitivity calculation would be 0/(0 + 3) = 0%, showing that three participants had strength asymmetries not identified by the jump asymmetry variable. The N/A value is a result of there being no true positives and no false negatives for the variable correlation. For example, the 1L-CMJ CMF asymmetry variable and hip abduction 20% threshold sensitivity calculation would be 0/(0 + 0) = 0%, showing that none of the 53 participants reached the asymmetry threshold for both the jump and strength measures, indicating no asymmetry was present in the participant pool.

## 4. Discussion

The current study investigated hip adductor, hip abductor, and eccentric hamstring strength metrics and asymmetries present in elite youth male soccer players. Additionally, using the investigated asymmetries, interest was specifically focused on whether countermovement jump asymmetry metrics could accurately detect hip abduction, hip adduction, and/or eccentric hamstring bilateral strength asymmetries. Through evaluating concentric impulse (CI), eccentric mean force (EMF), and concentric mean force (EMF) variables for the countermovement jumps and three strength assessments, the 1L-CMJ concentric impulse asymmetry variable was indicative of asymmetries present in the hip adduction strength variable at a 20% asymmetry threshold.

A key component to injury prevention is the ability to determine effective and efficient methods for detecting potential risk factors for future injury. The most commonly executed movements throughout sport, such as jumping, sprinting, and rapid change of direction, relay on muscular strength. Muscular strength can heavily influence the body’s ability to potentiate when using strength-power potentiation complexes [35]. The ability to potentiate earlier and to a greater extent is linked with the decrease in risk of injury. The force variables representing hip abductor, hip adductor, and eccentric hamstring strength were comparable to prior studies evaluating elite youth male athlete populations [36,37,38,39]. Additionally, muscle strength asymmetry proves to be an appropriate identifier and predictor of injury that can be utilized in the process [9,40]. However, most injury prevention and return to sport protocols do not include the assessment of muscle strength asymmetry [41,42]. The descriptive measures of the countermovement jump and strength asymmetry variables were consistent with prior studies of youth athletes [43,44,45,46]. The resulting study employed a commonly used countermovement jump (one-leg and two-leg) to identify whether the limb asymmetries detected in the jumps would provide insight to detectable strength asymmetries.

Secondarily, the current study investigated the ability of both the 2L-CMJ and 1L-CMJ asymmetry test variables to detect asymmetry compared to hip abduction, hip adduction, and eccentric hamstring strength asymmetry variables. The sensitivity and specificity statistical measures were used in order to assess the accuracy of the 2L-CMJ and 1L-CMJ asymmetry test variables to a reference standard [34,47]. A minimum cut-off threshold of 70% specificity [48,49,50,51] and 60% sensitivity [50,52,53] were put in place to ensure the inclusion of accurate variables based on how previous studies defined “good” sensitivity and specificity magnitudes. A test could have lower sensitivity and specificity values for a multitude of different reasons such as the CMJ variable having not been directly related to muscular strength. The variables representing the 2L-CMJ and 1L-CMJ in relation to hip abduction asymmetry at the 20% asymmetry threshold were not considered for the specificity threshold as zero participants reached the defined 20% asymmetry threshold.

For further investigation, the negative predictive value (NPV) best examined the predictability of the jump variable symmetry with respect to the examined strength symmetry variables in the studied population. Compared to sensitivity and specificity, positive and negative predictive values are clinically relevant as they consider the people being assessed [34]. The negative predictive value can predict how likely it is for someone to truly not have strength asymmetry present, in the case of not having asymmetry present in the corresponding countermovement jump variable [20]. Additionally, the NPV calculation includes the impact of the false negative value (asymmetry in the strength test despite symmetry in the CMJ variable) which we would want to see reduced in an ideal screening test. A minimally acceptable NPV value of 90% was set for variables to be considered further in analysis based on other studies considered to have resulted in a high and accepted NPV value [54,55,56,57].

Finally, the current study investigated whether the 1L-CMJ or 2L-CMJ jump variables that passed the accuracy thresholds were more suited to predict strength asymmetry in the observed population. At the 10% asymmetry threshold, no 1L-CMJ or 2L-CMJ asymmetry variables passed the 60% sensitivity and 70% specificity thresholds for detecting the strength asymmetries. However, all evaluated 2L-CMJ and 1L-CMJ variables reached an NPV magnitude of greater than 90% for hip abduction asymmetry. Despite the lower sensitivity values, it is important to note the potential function of the defined CMJ variables as a first step screening method for hip abduction asymmetry. Over 90% of the individuals with symmetry in the CMJ variables were also symmetrical in the hip abduction and hip adduction test; thus, in a screening scenario, the hip abduction test could be skipped in these individuals. At the 20% asymmetry threshold, only 1L-CMJ CI asymmetry in relation to hip adduction asymmetry reached the minimally acceptable NPV value of 90% with an NPV value of 97.9%, in addition to meeting the 70% specificity and sensitivity threshold values. As the 2L-CMJ CI asymmetry variable did not pass the sensitivity (0%) and specificity (96%) thresholds and only had a NPV value of 94.1%, the 1L-CMJ CI asymmetry variable is a better predictor of the presence of hip adduction asymmetry in youth male soccer players.

### Limitations

The current study has limitations that justify discussion. First, the present study only examined a limited range of variables in relation to the 2L-CMJ and 1L-CMJ, potentially limiting sensitivity and specificity detection abilities of the study. Future studies should include a broader range of variables to approach the sensitivity of the CMJ subtypes to provide a more all-encompassing analysis. Second, the study investigated a rather homogeneous cohort of skilled athletes with limited between-subject variability in relation to gender, skill level, sport type, and injury presence. The use of a uniform-like cohort results in the findings of the present study being largely applicable to only one population of athletes. Future studies should recruit a broader range of athletes that represent the population as a whole with more deviation. Additionally, what the data are not able to portray is the lack of experience or skill for the 1L-CMJ in the younger athletes. As many of the participants had never performed the jump skill before, many of them struggled with the task of balancing on one leg and jumping quickly. It should be considered in the future whether the lack of skill and difficulty with the jump for the population could account for some of the asymmetries detected. Expanding beyond the potential lack of experience of the youth athletes, it should be considered that there is an opportunity for the collected asymmetry values to have a high variability during the competitive season as well, presenting the single-day collection to be a limitation of the current study [58,59]. Another alteration future researchers should consider is the use of an individualized asymmetry threshold formula when evaluating a participant. García-García et al. found that the use of a specific asymmetry threshold formula classified more players as asymmetrical than with a fixed threshold as used in the current study [60]. Finally, the current study makes the assumption that asymmetry is abnormal within the examined population. However, it is possible and probable that some degree of asymmetry is not abnormal within the soccer population due to the kinematics and kinetic elements required of the sport. Future research should focus on a higher threshold for asymmetry as abnormal as the present study began to do by examining the twenty percent asymmetry threshold.

## 5. Conclusions

In conclusion, the findings of the current study suggest that the 1L-CMJ CI asymmetry at the 20% threshold is indicative of a youth male soccer player having hip adduction strength asymmetry. The concentric impulse variable for the 1L-CMJ has demonstrated more accuracy and applicability to the population of interest than the 2L-CMJ CI mirrored variable. It is important to note the limits of the current findings related to the small patient population size, and we encourage future research to further examine the discussed metrics with a larger and broader population. The use of the asymmetry of the concentric impulse variable during the 1L-CMJ test can potentially streamline the detection of hip adductor strength deficits by flagging players based on a quick jump test. This can improve the efficiency of the injury prevention testing process in youth male soccer players. Clinicians, coaches, and affiliated staff can then perform further hip testing and prescribe individualized training and strengthening routines on a player-by-player basis to correct detected asymmetries.

## Figures and Tables

**Table 1 sports-11-00077-t001:** Subject demographics.

	Total (*n* = 53)	
	Mean	±SD
Age (years)	14.2	1.3
Height (cm)	167.4	10.7
Mass (kg)	58.3	12.9

**Table 2 sports-11-00077-t002:** Measures of central tendency for each strength variable.

	Total (*n* = 53)	
	Mean (N)	±SD
Left Leg		
Hip AB AF	233.3	78.9
Hip AD AF	227.8	77.8
Hamstring AF	247.7	70.9
Right Leg		
Hip AB AF	237.5	79.9
Hip AD AF	237.9	80.6
Hamstring AF	254.5	80.6

AB: abduction, AD: adduction, AF: average force, N: Newtons.

**Table 3 sports-11-00077-t003:** Asymmetry of each variable.

Total (*n* = 53)			
	Mean	Median	SD
Hip AB AF	3.5%	3.9%	3.0%
Hip AD AF	6.0%	7.2%	5.0%
Hamstring AF	8.2%	10.0%	8.5%
2L-CMJ CI	5.0%	6.5%	5.6%
2L-CMJ EMF	11.3%	12.2%	7.8%
2L-CMJ CMF	5.0%	6.5%	5.7%
1L-CMJ CI	8.0%	9.7%	6.9%
1L-CMJ EMF	0.0%	0.0%	0.6%
1L-CMJ CMF	3.6%	5.1%	4.8%

CMJ: countermovement jump, CI: concentric impulse, EMF: eccentric mean force, CMF: concentric mean force, AB: abduction, AD: adduction, AF: average force, 2L: two legs, 1L: one leg.

**Table 4 sports-11-00077-t004:** Diagnostic accuracy at the 10% threshold.

	True Positive	False Positive	True Negative	False Negative
Hip Abduction Average Force				
2L-CMJ CI	1	9	41	2
2L-CMJ CMF	1	9	41	2
2L-CMJ EMF	1	30	20	2
1L-CMJ CI	1	21	29	2
1L-CMJ CMF	0	7	43	3
Hip Adduction Average Force				
2L-CMJ CI	2	8	34	9
2L-CMJ CMF	2	8	34	9
2L-CMJ EMF	7	24	18	4
1L-CMJ CI	5	17	25	6
1L-CMJ CMF	1	6	36	10
Nordic Hamstring Average Force				
2L-CMJ CI	5	5	25	18
2L-CMJ CMF	5	5	25	18
2L-CMJ EMF	13	18	12	10
1L-CMJ CI	10	12	18	13
1L-CMJ CMF	1	6	24	22

CMJ: countermovement jump, CI: concentric impulse, EMF: eccentric mean force, CMF: concentric mean force, AB: abduction, AD: adduction, 2L: two legs, 1L: one leg.

**Table 5 sports-11-00077-t005:** Diagnostic accuracy at the 20% threshold.

	True Positive	False Positive	True Negative	False Negative
Hip Abduction Average Force				
2L-CMJ CI	0	2	51	0
2L-CMJ CMF	0	2	51	0
2L-CMJ EMF	0	9	44	0
1L-CMJ CI	0	6	47	0
1L-CMJ CMF	0	1	52	0
Hip Adduction Average Force				
2L-CMJ CI	0	2	48	3
2L-CMJ CMF	0	2	48	3
2L-CMJ EMF	0	9	41	3
1L-CMJ CI	2	4	46	1
1L-CMJ CMF	0	1	49	3
Nordic Hamstring Average Force				
2L-CMJ CI	0	2	46	5
2L-CMJ CMF	0	2	46	5
2L-CMJ EMF	0	9	39	5
1L-CMJ CI	1	5	43	4
1L-CMJ CMF	0	1	47	5

CMJ: countermovement jump, CI: concentric impulse, EMF: eccentric mean force, CMF: concentric mean force, AB: abduction, AD: adduction, 2L: two legs, 1L: one leg.

**Table 6 sports-11-00077-t006:** Asymmetry evaluated at a 10% threshold.

	Se (%)	Sp (%)	PPV (%)	NPV (%)
Hip Abduction Average Force				
2L-CMJ CI	33.3	82	10	95.4
2L-CMJ CMF	33.3	82	10	95.4
2L-CMJ EMF	33.3	40	3.2	90.9
1L-CMJ CI	33.3	58	4.5	93.5
1L-CMJ CMF	0	86	0	93.5
Hip Adduction Average Force				
2L-CMJ CI	18.2	81	20	79.1
2L-CMJ CMF	18.2	81	20	79.1
2L-CMJ EMF	63.6	43	22.6	81.8
1L-CMJ CI	45.5	60	22.7	80.6
1L-CMJ CMF	9.1	86	14.3	78.1
Nordic Hamstring Average Force				
2L-CMJ CI	21.7	83	50	58.1
2L-CMJ CMF	21.7	83	50	58.1
2L-CMJ EMF	56.5	40	41.9	54.6
1L-CMJ CI	43.5	60	45.5	58.1
1L-CMJ CMF	4.4	80	14.3	52.2

CMJ: countermovement jump, CI: concentric impulse, EMF: eccentric mean force, CMF: concentric mean force, Se: sensitivity, Sp: Specificity, PPV: positive predictive values, NPV: negative predictive values, N/A: No asymmetry present, 2L: two legs, 1L: one leg.

**Table 7 sports-11-00077-t007:** Asymmetry evaluated at a 20% threshold.

	Se (%)	Sp (%)	PPV (%)	NPV (%)
Hip Abduction Average Force				
2L-CMJ CI	N/A	96.2	0	100
2L-CMJ CMF	N/A	96.2	0	100
2L-CMJ EMF	N/A	83.0	0	100
1L-CMJ CI	N/A	88.7	0	100
1L-CMJ CMF	N/A	98.1	0	100
Hip Adduction Average Force				
2L-CMJ CI	0	96	0	94.1
2L-CMJ CMF	0	96	0	94.1
2L-CMJ EMF	0	82	0	93.2
1L-CMJ CI	66.7	92	33.3	97.9
1L-CMJ CMF	0	98	0	94.2
Nordic Hamstring Average Force				
2L-CMJ CI	0	95.8	0	90.2
2L-CMJ CMF	0	95.8	0	90.2
2L-CMJ EMF	0	81.3	0	88.6
1L-CMJ CI	20	89.6	16.7	91.5
1L-CMJ CMF	0	97.9	0	90.4

CMJ: countermovement jump, CI: concentric impulse, EMF: eccentric mean force, CMF: concentric mean force, Se: sensitivity, Sp: Specificity, PPV: positive predictive values, NPV: negative predictive values, N/A: No asymmetry present, 2L: two legs, 1L: one leg.

## Data Availability

The data presented in this study are available upon request from the corresponding author.

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
