# Peer review of "Ability of Countermovement Jumps to Detect Bilateral Asymmetry in Hip and Knee Strength in Elite Youth Soccer Players"

_sports, 2023, doi:10.3390/sports11040077_

Round 1
Reviewer 1 Report
Basic reporting
The authors reported hip abductor, and eccentric hamstring strength metrics and asymmetries present in elite youth male soccer players. Whilst the study undoubtedly has merit, there are some aspects that need clarification, to improve the readability of the manuscript.
KEYWORDS
· Please, do not repeat in keywords the tittle words
· Add “football” as a keyword.
INTRODUCTION
General comment: Introduction section is to short. Authors should
· L 39-42: Which sports? Contact sports? Better compare with team sports.
· L 50: Add reference.
· L 59-61: I suggest adding the following reference
o “Fort-Vanmeerhaeghe, A.; Mila-Villarroel, R.; Pujol, M.; Arboix-Alió, J.; Bishop, C. Higher vertical jumping asymmetries and lower physical performance are indicators of increased injury incidence in youth team-sport athletes. J. Strength Cond. Res. 2020.
· L 62: I think it would be interesting to add a paragraph about the realatisonship between inter-limb asymmetries and performance in differen team sport athletes. It is a well-documented topic that taletly has been reported in many investigations. I suggest introducing the following references:
o Madruga-Parera, M.; Bishop, C.; Beato, M.; Fort-Vanmeerhaeghe, A.; Gonzalo-Skok, O.; Romero-Rodríguez, D. Relationship between interlimb asymmetries and speed and change of direction speed in youth handball players. J. Strength Cond. Res. 2019, 35, 3482–3490.
o Arboix-Alió, J.; Buscà, B.; Busquets, A.; Aguilera-Castells, J.; de Pablo, B.; Montalvo, A.M.; Fort-Vanmeerhaeghe, A. Relationship between Inter-Limb asymmetries and physical performance in rink hockey players. Symmetry 2020, 12, 2035.
o Bishop, C.; Read, P.; McCubbine, J.; Turner, A. Vertical and horizontal asymmetries are related to slower sprinting and jump performance in elite youth female soccer players. J. Strength Cond. Res. 2018, 35, 56–63.
MATERIAL AND METHODS
Participants
· Please introduce the sample’s training volume per week (hours x week).
· Moreover, I suggest you improve the description of the sample selection. Please, could you deeply present how you proceed to select your sample?
It is a representative sample? If is it, introduce please the Sample size calculation.
· Which is the samples’ PHV?
Procedures
General comment: Divide the tests used into subheadings in order to help readers comprenhension.
· Please present information about how and where tests were performed (kind of warm-up performed, who organized and applied the test, etc.). Please detail all the main information and the necessary details in order to provide the reader with a clear picture of how tests were performed.
· Were the test randomized? Which procedure did you use to randomize the order?
Statistical Analysis
· L 141: Introduce the reference for the ASI formula.
· L 149-157: This information it is not necessary. It can be delated.
· The authors should also provide reliability analyses of the tests because they measured adolescents who do not have much experience in fitness tests.
RESULTS
General comment: There are to many tables. Please reduce it.
· Table 1 is not necessary. This information should be introduced into the sample description section.
· Table 2: Please provide the intraclass correlation coefficients (ICC) and CV in your measured variables.
DISCUSSION
L 393: Since the asymmetry value has a high variability during the competitive season, the fact of measuring it on a single day is a clear limitation. Authors should include this limitation:
o Fort-Vanmeerhaeghe, A.; Bishop, C.; Buscà, B.; Vicens-Bordas, J.; Arboix-Alió, J. Seasonal variation of inter-limb jumping asymmetries in youth team-sport athletes. J. Sports Sci. 2021, 39, 2850–2858.
o Bishop, C., Read, P., Chavda, S., Jarvis, P., Brazier, J., Bromley, T., & Turner, A. (2020). Magnitude or direction? Seasonal Variation of interlimb asymmetry in elite academy soccer players. Journal of Strength and Conditioning Research, 1. Doi:10.1519/JSC.0000000000003565
CONCLUSIONS
The conclusion section should be further developed. Moreover, how these findings can help S&C coaches and practitioners? Which practical implications can it have in training sessions?
Reviewer 2 Report
It is a well-written manuscript. I enjoyed reading it. However, I have a few minor comments
What were the inclusion and exclusion criteria? Did all the participants were healthy?
How were the athletes recruited?
The result and discussion are written well.
Round 2
Reviewer 1 Report
The authors have modified the text appropriately in response to the requests I had made.
In my opinion, the article is now much more understandable and I do not have more suggestions to improve the quality of the present research.